# Adherence to High Dietary Diversity and Incident Cognitive Impairment for the Oldest-Old: A Community-Based, Nationwide Cohort Study

**DOI:** 10.3390/nu14214530

**Published:** 2022-10-27

**Authors:** Yangyang Song, Lu Zeng, Julin Gao, Lei Chen, Chuanhui Sun, Mengyao Yan, Mengnan Li, Hongli Jiang

**Affiliations:** 1Dialysis Department of Nephrology Hospital, The First Affiliated Hospital of Xi’an Jiaotong University, Xi’an 710061, China; 2The First Affiliated Hospital of Xinxiang Medical University, Xinxiang 453100, China

**Keywords:** cognitive decline, the oldest-old, Chinese, dietary diversity changes, human longevity

## Abstract

Background and aims: Dietary diversity change is associated with cognitive function, however, whether the effect still exists among the oldest-old (80+) is unclear. Our aim was to examine the effect of dietary diversity changes on cognitive impairment for the oldest-old in a large prospective cohort. Methods: Within the Chinese Longitudinal Healthy Longevity Study, 6237 adults older than 80 years were included. The dietary diversity score (DDS) was assessed by a simplified food frequency questionnaire (FFQ). Cognitive impairment was defined as a Mini-Mental State Examination (MMSE) score lower than 18 points. Cognitive decline was defined as a reduction of total MMSE score ≥3 points, and cognitive decline of different subdomains was defined as a reduction of ≥15% in the corresponding cognitive domain. The multivariate-adjusted Cox proportional hazard model evaluated the effects of DDS change on cognitive decline. The linear mixed-effect model was used to test subsequent changes in MMSE over the years. Results: During 32,813 person-years of follow-up, 1829 participants developed cognitive impairment. Relative to the high–high DDS change pattern, participants in the low–low and high–low patterns were associated with an increased risk of cognitive impairment with a hazard ratio (95% confidential interval, CI) of 1.43 (1.25, 1.63) and 1.44 (1.24, 1.67), and a faster decline in the MMSE score over the follow-up year. Participants with the low–high pattern had a similar incidence of cognitive impairment with HRs (95% CI) of 1.03 (0.88, 1.20). Compared with the stable DDS status group (−1–1), the risk of cognitive impairment was higher for those with large declines in DDS (≤−5) and the HR was 1.70 (95% CI: 1.44, 2.01). Conclusions: Even for people older than 80, dietary diversity change is a simple method to identify those who had a high risk of cognitive decline. Keeping high dietary diversity is beneficial for cognitive function and its subdomain even in the final phase of life, especially for females and the illiterate oldest-old.

## 1. Introduction

Age-related cognitive impairment has become a significant public health challenge. The rapid population aging is expected to lead to an increase of 75 million old people with dementia worldwide by 2030 [1]. In China, 16.9% of elderly people aged over 80 years old suffer from mild cognitive impairment [2]. As no effective pharmacologic therapies for cognitive function are available yet, identifying potentially modifiable risk factors is critical in preventing cognitive decline for the elderly. A high diet quality is considered as a critical protective factor in cognitive function, and the beneficial effect is more pronounced for people aged 80 years or older [3,4].

Most previous studies have focused on the static diet quality of older people, which have ignored dynamic changes in the diet quality over time. However, the oldest-old are susceptible to dietary diversity change due to loss of appetite, degeneration of the digestive system, and lower economic status. A long-term cohort of 17,959 older people showed that only 32.92% of them kept diet diversity at stable status for 2–3 years [5]. Therefore, some measurement errors could occur if the changing trends in diet quality are not taken into account. Focusing on the cognitive impairment risk associated with the diet changes, not just static diet quality is crucial for the oldest-old since their diet usually changes in reality. 

Evidence has shown the association between overall diet quality change and cognitive function among younger-old adults based on the Mediterranean diet (Me-DI), the Dietary Approaches to Stop Hypertension (DASH) diet, and the Mediterranean-DASH Intervention for Neurodegenerative Delay (MIND) diet. Nevertheless, their results have been controversial. Some studies have reported that higher adherence to specific dietary patterns promoted better cognitive function [6,7], whereas other cohorts could not replicate the association [8,9,10]. Dietary culture, behavior, and age distribution variations can explain these discrepancies [11]. The Western style dietary patterns (Me-DI, DASH, MIND) may not be suitable for other regions. Limited studies have applied these specific dietary patterns among Chinese people [6]. Meanwhile, the small number of the oldest-old and short follow-up time cannot reflect the long-term effects of diet changes. 

The Dietary Diversity Score (DDS) has been well-recognized as an essential tool to assess diet quality. The DDS is considered as various types of food groups in accordance with the local dietary guideline [12,13]. Compared with the complicated measurement of other dietary patterns, the DDS is easier and more straightforward to complete for the oldest-old people and is applicable to Chinese dietary culture [14,15,16]. Previous researchers have proven the beneficial effect of high DDS on cognitive function [16,17,18]. However, little is known about whether the association still exists for those in the final phase of life-span. The most related cohort [19], which explored the relationship between dietary diversity change and cognitive function, found that keeping higher dietary diversity could reduce the risk of cognitive impairment among participants older than 65 years old. Our aim was to extend the findings of this cohort [19] in three ways. First, our cohort focused not only on binary outcomes (e.g., cognitive impairment vs. no cognitive impairment), but on incident cognitive decline, which can detect subtler changes in cognitive function. Second, we will explore the longitudinal changes of cognitive function in DDS change patterns over long-term follow-up. Third, we will examine whether the DDS changes had an effect on different cognitive domains of cognitive function. Furthermore, we focused on the oldest-old (80+) rather than people aged over 65 years old. Based on the Chinese Longitudinal Healthy Longevity Study (CLHLS), our study aimed to capture the effects of dynamic features of DDS change patterns on cognitive function among adults over 80 years old.

## 2. Materials and Methods

### 2.1. Data Source

Our study was performed based on the Chinese Longitudinal Healthy Longevity Study (CLHLS). CLHLS is a longitudinal, national prospective cohort for old adults. It used a multi-stage stratified random sampling method in 866 different counties and cities of 23 provinces in China. Moreover, the centenarians from the 801 cities or counties were selected randomly in the whole country. People of predefined sex and age living nearby were randomly invited based on the centenarians’ code numbers [20] to match with the centenarians. The cohort can cover nearly 85% of old Chinese people. Therefore, CLHLS can be used as representative data of the oldest-old to explore the determinants of longevity [21,22]. For more details about the data quality of the CLHLS, the readers are referred to previous studies [20,21]. The oldest-old (over 80 years old) were recruited in a wave between 1998 and 2000. Moreover, the old adults (over 65 years old) were enrolled from 2002. New participants were recruited every two to four years. Trained interviewers conducted a structured questionnaire interview (diet, lifestyle, and medical history) for each participant in their home. All old adults or their representatives signed written consent forms. The Ethics Committee approved the CLHLS of Peking University (IRB00001052-13074). 

We focused on participants above 80 years old from two successive cohorts (20 years: 1998–2018; 18 years: 2000–2018) within the CLHLS. It included participants who were free of cognitive impairment (MMSE score < 18) at the baseline, and they needed to have the baseline and first follow-up of the DDS and MMSE scores. People who were younger than 80 years old were excluded. We included the longest follow-up period of data for the final analysis when participants were present in two cohorts. Finally, our study included 6237 participants after excluding 4833 duplicated participants (Appendix A).

### 2.2. Measurement of DDS

Participants were required to complete a simplified food frequency questionnaire in a face-to-face review. The validity and scientificity of the food frequency questionnaire have been verified in previous literature [23,24], especially for the oldest-old [18]. The information about nine major food groups (fresh vegetables, food made from beans, fish, fresh fruit, tea, garlic, meat, eggs, and preserved vegetables) was collected. The trained interviewer asked: “How often do you have the food group at present?”. The answers were classified as often (more than five times a week), occasionally (1–4 times a week), or rarely (less than once a week). Then, we defined the corresponding score as 0 (rarely), 1 (occasionally), and 2 (often). The total DDS score (0 to 18) was the sum score of the above nine food groups. The score of plant-based DDS (0–12) was the sum score of fresh vegetables, garlic, food made from beans, preserved vegetables, fresh fruit, and tea. The animal-based DDS (0–6) was the sum score of eggs, meat, and fish. 

### 2.3. Measurement of DDS Change Patterns

We calculated two types of DDS change patterns within the first follow-up period. The Food and Agriculture Organization of the United Nations dietary diversity guidelines recommend that DDS could be classified based on the mean value. The validity and reproducibility of the DDS change pattern for Chinese old adults have been described in previous studies [5,19]. Evidence shows that the method of DDS change pattern could better categorize dietary diversity change, and the “high-high DDS change pattern” had the lowest mortality risk and cognitive impairment for adults over 60 years old [5,19]. The first DDS change pattern was operationalized as follows. (1) Categorized baseline DDS into two groups: the low group (0–9 points) and the high group (10–18 points); classified the planted-based DDS into two groups: the low (0–6 points) and the high (7–12 points) group; animal-based DDS were divided into two groups: the low (0–3 points), the high (4–6 points). (2) Calculated a new variable to represent the changing pattern from the baseline and created the first follow-up including the high–high, high–low, low–high, and low–low DDS change patterns.

The second measurement of the DDS change pattern was calculated by the first-2 year DDS minus the baseline DDS, then classified into five groups based on Cox models with penalized splines [25,26] including a large decline (DDS change score ≤ −5), moderate decline (−4 ≤ DDS change score ≤ −2), stable status (−1 ≤ DDS change score ≤ 1), moderate improvement (2 ≤ DDS change score ≤ 4), and large improvement (DDS change score ≥ 5). 

### 2.4. Assessment of Cognitive Function

A trained interviewer evaluated cognitive function through the Chinese version of the MMSE scale, and the score range was 0–30. A higher MMSE score means a better performance of cognitive function. Several studies have verified the reliability and validity of the Chinese MMSE for older people [20,27]. According to previous studies, we regarded the response (unable to answer) of the oldest-old as “wrong” [28]. Since nearly half of the included oldest-old in the cohort were illiterate, we defined cognitive impairment as a MMSE score lower than 18 [14,29]. The five subdomains of cognitive function were evaluated: orientation, registration, attention and calculation, memory, language and visuospatial ability (Appendix A) [30]. We also evaluated the incident cognitive decline at the exit visit as secondary outcomes: (1) total MMSE score reduced ≥3 points, and (2) cognitive function score reduced by 15% for subdomains of cognitive function [30,31].

### 2.5. Covariates

The following baseline variables were adjusted as covariates: sociodemographic confounders included age (years), sex, body mass index (BMI, calculated as weight in kilograms divided by height in meters squared), number of teeth, use of artificial dentures (yes or no), occupation, marital status, residence type (urban or rural), education level (illiteracy and literacy), and living pattern (live with a family member or alone/in a nursing home). Health behavior included regular exercise (yes or no), tobacco smoking (current smoker, former smoker, never smoking), activities of daily living (ADL), and drinking status (current drinker, former drinker, never drinking). Self-reported chronic diseases included hearing disorders, diabetes, hypertension, digestive system diseases, cerebrovascular diseases, cancer, eye diseases, and respiratory diseases. The ADL measurement tool included six essential tasks related to independent individual life: eating, toileting, bathing, dressing, indoor activities, and continence [32]. The score of each item was zero if the participants could not perform the task independently, and “1” means that they could complete the task by themselves.

### 2.6. Statistical Analysis

The characteristics of the included oldest-old with different DDS change patterns were analyzed. The mean (SD) and number (percentage) were used to present the continuous variables and categorical variables. The different subgroups of DDS change score were compared by the independent samples *t*-test or one-way analysis of variance. A post-hoc analysis was conducted by the least significance difference (LSD) test.

The Cox proportional hazard models were used to explore the relationship between two types of DDS change patterns and cognitive impairment. The time when cognitive impairment first occurred was regarded as the endpoint. The follow-up period was calculated from the baseline to whichever occurred first: the first occurrence of cognitive impairment, death, loss to follow-up, or the endpoint in the cohort. The percentage of missing values of covariates was less than 2%, and we performed the multiple imputation method to rectify missing covariate values [33]. The proportional hazard assumption of categorized and continuous variables was satisfied by the Kaplan–Meier curves and linear test regression of scaled Schoenfeld residuals on time functions. Restricted cubic spline analysis was performed to analyze whether non-linear relationships existed between the DDS change scores (as a continuous variable) and cognitive impairment [34]. 

For the DDS change pattern, the adjustment was accomplished by two models: (1) model 1, which adjusted for age and sex; (2) model 2, which additionally adjusted for the number of teeth, BMI, occupation, marital status, use of artificial dentures, residence type, education level, and living pattern. Health behaviors included smoking status, ADL score, regular exercise, hear disorder, drinking status, diabetes, hypertension, digestive system diseases, cerebrovascular diseases and cancer, eye diseases, respiratory diseases, and baseline MMSE. Notably, we added the baseline DDS score as a potential confounder for analyzing the second DDS change pattern.

The MMSE score was evaluated for participants at the baseline and each follow-up time. We used the multilevel linear mixed-effects model to test the relationships of the DDS change patterns (the high–high, the high–low, the low–high, the low–low) with repeated measurements of the MMSE change score [35]. The associations of animal-based and plant-based DDS change patterns with cognitive function were also explored in the subsequent year. The fixed effect contained the DDS change pattern, follow-up time, and their interaction. The random effect included random intercept and slope for time, age, sex, number of teeth, BMI, occupation, marital status, artificial dentures, residence type, education level, and living pattern. Health behaviors included smoking status, ADL score, regular exercise, hearing disorder, drinking status, diabetes, hypertension, digestive system diseases, cerebrovascular diseases and cancer, eye diseases, respiratory diseases, and the baseline MMSE scores were adjusted in the model.

In addition, subgroup analyses were conducted to explore the relationship between cognitive impairment and DDS change patterns by different age groups (80–89 or over 90 years), sex, residence (urban or rural), living patterns (living with family or not), smoking status, drinking status, education level (illiterate or not), regular exercise (yes or no). To assess the potential effect modifications, we also performed a cross product of subgroup variables with DDS change patterns in the multi-variable model.

Sensitivity analysis was performed to determine whether the above results were robust. (1) Participants were excluded who had died or lost contact in the second follow-up since they might have been more prone to cognitive impairment; (2) we conducted a sensitivity analysis by using the definition of cognitive impairment (MMSE score less than 24); (3) participants with chronic diseases (hypertension, diabetes, hearing disorder, cancer, cerebrovascular diseases) at the baseline were excluded to minimize the potential reverse causation.; and (4) education was treated as a continuous variable to reduce the potential effect of education level further. We considered it statistically significant when a two-tailed *p* value was less than 0.05. All analyses were conducted by Stata SE 15.0.

## 3. Result

### 3.1. Participant Characteristics

Among the 6237 participants, 53.6% were female, 24.0% were married, and the mean age (standard deviation, SD) was 88.6 (7.0) years old at thee baseline. In terms of the baseline MMSE score and DDS score, the mean (SD) was 26.32 (3.3) and 9.2 (2.8) points, respectively. The percentage of high–high, high–low, low–high, and low–low DDS change pattern at the baseline was 27.5%, 17.4%, 20.2%, and 35.0%, respectively. More details about the basic characteristics are presented in Table 1. The participants with a low–low pattern were more likely to be female, illiterate, not in a marriage, live in rural areas, and do less exercise.

The mean follow-up period was 5.26 years, ranging from 1.42 to 20.0 years. As shown in Table 1, during the 32,813 person-years of follow-up, 1829 participants (29.32%) developed cognitive impairment. The incidence was 5.57 per 100 person-years (3.65, 6.30, 4.69, and 7.55 per 100 person-years for participants with high–high, high–low, low–high, and low–low DDS change patterns, respectively).

### 3.2. Association of DDS Change Patterns with Cognitive Impairment

We found that participants with high–high DDS change patterns suffered the lowest risk of cognitive impairment. In comparison with the oldest-old in the high–high DDS change group, there was a similar risk of cognitive impairment in the low–high DDS change pattern, and those in the high–low and the low–low DDS change pattern that suffered higher risks of cognitive impairment with the HRs were 1.44 (95% CI: 1.24, 1.67) and 1.43 (95% CI: 1.25, 1.63), respectively. Similar effects existed in the plant-based DDS change patterns and animal-based DDS change patterns on cognitive function (Table 2). Relative to the high–high group, the estimates for the high–low and low–low group of the planted-based DDS were 1.43 (95% CI: 1.23–1.67) and 1.44 (95% CI: 1.26–1.65), respectively. For the animal-based DDS, the estimates of HRs were 1.18 (95% CI: 1.00–1.39) and 1.22 (95% CI: 1.07, 1.40), respectively. More details are present in Table 2.

In the subgroup analyses (Table 3), when participants were stratified by age group, sex, smoking status, education level, residence type, drinking status, and living pattern, exercise, the results were similar to our main results. We found that the DDS pattern (high–low, low–low) had a significantly higher risk of cognitive impairment than the high–high pattern. The univariate model is shown in Appendix A. Additionally, the negative effect of the low–low DDS pattern on cognitive function decreased with increments in age. Moreover, females were more easily affected by DDS change. The adjusted HR associated with the DDS change pattern (high–low, low–low) versus the high–high pattern was 1.45 (95% CI: 1.20–1.76), 1.44 (95% CI: 1.22–1.71) for the female, and 1.40 (95% CI: 1.09–1.77), 1.37 (95% CI: 1.10–1.70) for the male. 

We found that low–low and high–low DDS change pattern was associated with higher hazards of cognitive decline including orientation, registration, attention and calculation, language, and visuospatial abilities (Table 4). Similar relations were also found in participants with plant low–low DDS change patterns. As for memory, only the high–low total DDS change pattern was associated with faster cognitive decline (Table 4). Only animal DDS change patterns (high–low, low–low) were associated with a faster decline in attention and calculation, language, and visuospatial abilities.

In multi-adjusted linear mixed-effects models, participants with low–low, high–low, and low–high DDS change patterns experienced a faster decline in annual global cognitive function than high–high DDS change patterns (Table 5, Figure 1). Moreover, participants with either plant or animal low-low DDS change patterns had a rapid decline rate in global cognition than those with the high–high DDS pattern over the follow-up (Appendix A, Figure 1). 

### 3.3. DDS Change Score and Cognitive Impairment

Among the whole cohort, the percentage of large decline, small decline, stable status, small improvement, and large improvement was 9.4%, 20.5%, 36.4%, 23.8%, and 9.9%, respectively (Appendix A). The incidence of cognitive impairment for a large decline, small decline, stable status, small improvement, and large improvement was 7.51, 6.23, 5.45, 4.8, and 4.98 per 100 person-years, respectively.

When we assessed the DDS change scores (continuous variable) and cognitive function by restricted-cubic-spline analyses, we identified a reverse-J relationship between the DDS change score with cognitive impairment (*p* = 0.028, Figure 2). Relative to participants whose DDS change remained stable, those with a large and small decline in DDS was associated with a higher risk of cognitive impairment with the HRs were 1.70 (95% CI: 1.44, 2.01) and 1.23 (95% CI: 1.08, 1.40) while the oldest-old with large improvement or small improvement in DDS had a lower risk with the HRs was 0.83 (95% CI: 0.73, 0.94) and 0.75 (95% CI: 0.63, 0.90) (Table 6).

Similar findings across age, gender, and education level were found for DDS change in the stratified analysis (Appendix A). Compared with stable status, large declines in DDS resulted in a significantly higher risk of cognitive impairment in all subgroups. Additionally, a small decline in DDS led to a significantly increased risk of cognitive impairment in the participants aged 80–89 years old, the illiterate group, and the female group. Nevertheless, it was not shown in the participants over 90 years old, and this result might be explained by the survival bias.

### 3.4. Sensitivity Analyses 

Sensitivity analyses showed a similar result. We excluded participants who had died or lost contact in the second follow-up, and using the definition of cognitive impairment (MMSE score less than 24); the participants with chronic diseases at the baseline were also excluded. We considered education as a continuous variable and excluded self-reported cerebrovascular diseases (Appendix A). Furthermore, both small and large improvements in the DDS scores had a protective effect in the 90+-year-old and female subgroup. Sensitivity analyses were performed, and there were no material changes in the results (Appendix A).

## 4. Discussion

In this community-based prospective cohort study, we found that compared with those participants maintaining high DDS, those with low–low DDS change patterns had an increased risk of cognitive impairment and were associated with steeper global cognition decline during 20 years of follow-up. In addition, compared with those in stable DDS status, the oldest-old who had large declines in DDS score within two years was associated with a significantly increased risk of cognitive impairment.

Diet diversity was related to lower cognitive decline among the oldest-old, as proven in previous studies. Many studies have explored the Western dietary pattern, which is not applicable to old Chinese people [11,36,37]. To date, only one long-term study has comprehensively highlighted the effect of dietary diversity on cognitive function for the Chinese oldest-old. In their study, Zheng et al. [18] demonstrated that participants with higher baseline DDS scores had a lower risk of cognitive impairment than those with lower DDS. They also found that a higher DDS score could attenuate the rate of cognitive decline during long-term follow-up. However, their study focused on static dietary diversity status; the dynamic features of DDS that reflect the change of nutrient adequacy were ignored. Notably, accumulating evidence proved that DDS change was associated with mortalities among the oldest-old [5,38,39], but the relationship between DDS change and cognitive function was unclear. The current study aimed to explore the relationship between the DDS change patterns and cognitive function among the oldest-old over long-time follow-up. Our finding expands the results of Zheng et al. [18] by demonstrating that maintaining high DDS or improving DDS could lower the risk of cognitive impairment regardless of baseline DDS. 

In accordance with previous studies, we evaluated the effect of adherence to healthy dietary patterns on cognitive function. Uniquely, this research focused on the oldest-old ignored in previous studies and assessed the DDS change trends that have not been previously explored. Our results were consistent with some longitudinal cohorts where the high-adherence diet patterns had better cognitive function for older people. A 7-year cohort of Greek elderly participants aged 65 years or older suggested that those keeping the Me-DI diet had less cognitive function decline [40]. However, other cohorts found that the association did not exist in Australians aged 60–64 years (*n* = 1528) [9] and the French aged over 65 years old (*n* = 1410) within 2.2–12 years of follow-up [41]. Notably, the positive effect was replicated among participants near the Mediterranean basin [42], but limited or null associations existed for people living in non-Mediterranean regions [43]. Furthermore, the variation in different dietary cultures undermines its generalizability, as some Western style diets (the dietary products, soft drinks) are not consumed among the Chinese oldest-old. The researchers also verified that the protective effects of diet diversity were geographically generalizable and age-specific [11]. Since there is no standardized measurement for diet diversity, the optimal measurement of diet diversity is critical to assess its effect on cognitive function. Our findings confirm that it is scientific to choose these nine common food groups.

Our results showed that participants with high–high DDS patterns showed the minimum downtrend in MMSE scores, followed by the low–high DDS change pattern. Participants with high–low DDS change patterns had accelerated declines in cognitive function over the follow-up. The result is consistent with the previous study; compared to participants with high–high DDS patterns, those with low–medium still had a higher cognitive impairment risk (HR 2.30, 95% CI 1.90–2.78) for adults over 60 years old [19]. It should be noted that improving the DDS score from a low score to a high score still had a higher risk of cognitive function decline than keeping a high DDS score. Thus, it is beneficial to keep a high and stable DDS score; it is important to promote diet diversity from early old age to prevent cognitive decline. We also found that compared with the low–low DDS pattern, participants with low–high was associated with a lower decline in cognitive function. Meanwhile, our results demonstrated that there was no significant difference in incident cognitive impairment between low–high DDS change pattern with a high–high DDS change pattern (Table 2), and compared with the stable status of low DDS, even a small improvement in DDS could reduce the risk of cognitive impairment (HR = 0.83, 95% CI: 0.73,0.94). Therefore, our findings suggest that improving DDS is helpful to reduce the incidence of cognitive impairment for those with a low DDS score.

Additionally, the protective effects for cognitive function were observed for adherence to the high animal-based DDS and plant-based DDS change pattern. One possible mechanism might be that the brain of the oldest-old is more likely to show oxidative damage [44]. Some diet components and the synergistic effects of different food groups might have an anti-inflammatory and antioxidant effect on the brain [45,46], affecting neuronal pathways and physiological mediators [47,48]. This means that maintaining low dietary diversity causes severe oxidative damage [49].

The beneficial effect of improving dietary diversity on cognitive function has been confirmed even in the final phase of life. In the current study, compared with participants with a “low–low” DDS change pattern, the incidence of cognitive impairment of “low–high” decreased by 28 per 1000 person-years for people over 80 years old. An interesting finding was that the “low–high” had a significant rapid cognitive decline (*p* < 0.05) relative to the “high–high” DDS change pattern, but the incidence of cognitive impairment was similar to that of the “high–high” DDS change pattern.

In terms of the DDS change trend, our findings suggested that large declines in the DDS change scores resulted in a higher risk of cognitive impairment among the oldest-old. Evidence shows that an extreme decline in DDS exerts a severe effect on cardiovascular diseases, cancer, and overall mortality [50,51]. A longitudinal cohort of 12,974 older people found that relative to those with stable DDS status, others with large declines in DDS faced a higher mortality risk (HR: 1.15, 95% CI: 1.09–1.22) [5]. One possible mechanism may be that dietary diversity decline was related to microbiome stability [52]. A significant change in DDS may alter microbiome composition and then impact the cognitive performance of the brain–gut–microbiome axis [53,54]. In addition, the relationship between DDS change and cognitive impairment showed a reverse J-shape in this study. Therefore, it is critical to treat the history of large declines in DDS change as a potential risk factor to predict cognitive impairment for the oldest-old.

Another interesting finding was that female and illiterate participants were more easily affected by DDS change. A large or small decline could increase the risk of cognitive impairment, but a small or large improvement could attenuate the risk. The association was not found in other subgroups. In line with our research, a cross-sectional study proved that higher DDS was associated with reduced risks of cognitive impairment in older Chinese women [55]. Similarly, another study found that mild cognitive impairment was inversely related to the dietary pattern scores [56], especially for women. Moreover, the CLHLS showed that malnutrition contributed to a worse cognitive function, especially for the oldest illiterate females. The dietary intake’s bio-physiological sensitivity may account for the gender difference [57,58]. Participants with high educational levels have higher cognitive reserves (CR), which was considered as a compensation mechanism for the same brain damage on cognitive function [59]. Our results emphasize that the oldest-old with illiterate or females need to be paid more attention to in clinical practice.

The major strength of this study is the national population-based representative cohort for the oldest-old in China [22]. The long-term and repeated follow-up, well-designed cohort allowed us to explore the long-term relation of DDS dynamic change with cognitive function. To our knowledge, the current study is the first to investigate the association between DDS change and cognitive impairment for the oldest-old in China. The average age of the included participants was 88.6 years old. These populations more easily suffer from dementia, and our study may offer some evidence for its prevention. In the oldest-old, the effect of DDS might be overestimated for many reasons. Participants with high–high DDS change patterns might have a better cognitive function at baseline, and some chronic disease deterioration might result in a large decline in DDS. In this study, we adjusted these confounders including the baseline MMSE, some chronic diseases, ADL score, and BMI. We also conducted sensitivity analyses to confirm the robustness of our results. 

Our study still has several limitations: first, the detailed dietary intake was not quantitative because the current cohort design was a preliminary study, so we were unable to adjust for energy intake in the analyses. Since energy intake largely depends on the oldest-old, some important variables such as age, sex, BMI, comorbidities, economic status, exercise and lifestyle were adjusted in our analyses. Although many studies have demonstrated that this DDS change pattern is associated with health outcomes among older people [5,19], the current method to evaluate DDS change without quantitative data is not a gold standard, which limits its generalizability to other populations. Future studies need to further confirm the validity of the DDS change pattern. Second, although we chose the optimal food groups, nuts and milk were not included because most of the participants could not afford them; third, the confounding factors were based on self-reported data, which may cause recall bias. Furthermore, most of the participants (81.5%) lived with family members, and the DDS change risk of those who lived alone or lived in nursing homes might have been underestimated. Fourth, half of the oldest-old only received less than one year of education, which may limit its application to other older people with high education levels. Therefore, caution should be taken when explaining the cause–effect relationship between DDS change and cognitive function due to the observational nature of the current study. However, considering that worse cognitive function might influence the diet diversity, the exclusion of patients with poor cognitive function (MMSE < 18) might minimize the bias. Notably, the different cognitive domains in the current study were calculated based on the MMSE items. The method has been used in previous studies [30], and the MMSE domain-specific cognitive impairment aligns with the performance in detailed neuropsychological tests, which might be useful to guide further neuropsychological tests [60,61]. It can be considered as a proxy since the assessments of domain-specific cognitive function in our study were unavailable. However, it might still be poorly informative. Future studies should examine the association between DDS change and specific cognitive function through comprehensive and detailed tests. Finally, the present study only included people with normal cognitive performance at baseline. Therefore, further studies are needed to validate the generalizability of participants with cognitive impairment.

## 5. Conclusions

In the large national, representative longitudinal cohort, the oldest-old, keeping low–low and significant declines in total DDS and plant-based change patterns had a higher risk of cognitive impairment and cognitive decline in total cognitive function and its subdomain except for the memory domain. The animal-based DDS change was associated with cognitive subdomains such as attention and calculation, language, and visuospatial. The low–low total DDS change pattern had the highest subsequent cognitive decline over long-term follow-up. Our results support that dynamic DDS change scores might be a potential marker of cognitive impairment in the final life span phase, especially for females and illiterate people. Therefore, researchers should focus on improving the DDS in clinical practice to protect cognitive function at a younger age and maintain high–high DDS change patterns for the oldest-old.

## Figures and Tables

**Figure 1 nutrients-14-04530-f001:**
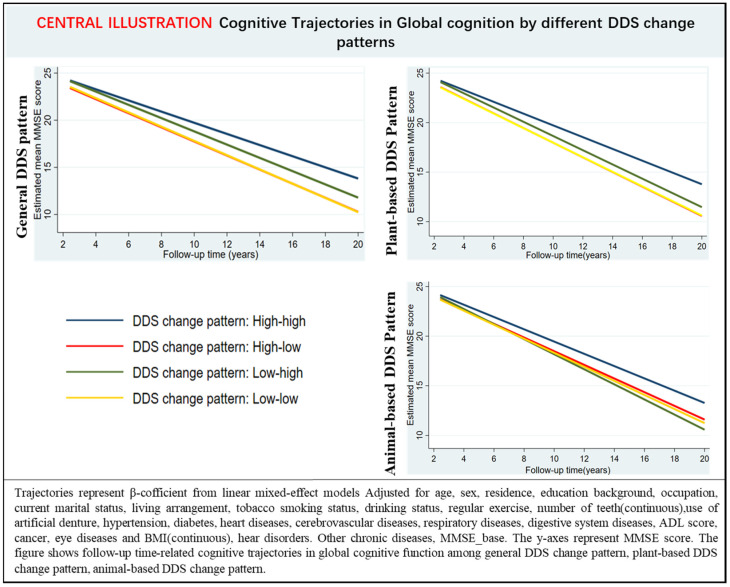
Estimated mean score of the mini-mental state examination with 95% CIs at follow-up year intervals among participants with different DDS change patterns.

**Figure 2 nutrients-14-04530-f002:**
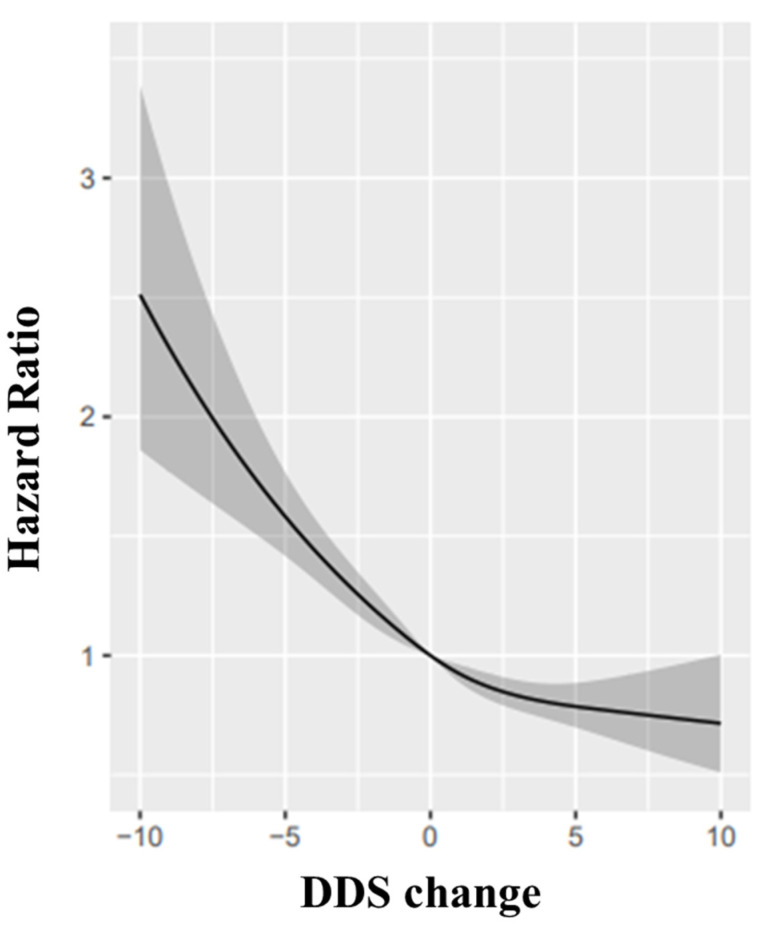
Restricted cubic splines for the association of DDS change with cognitive impairment: the reference point is the median value of DDS change (0), with knots placed at 10th, 50th, and 90th percentiles, after adjusting for age, sex, BMI, the number of teeth, use of artificial dentures, occupation, marital status, residence type, education level, living pattern; tobacco smoking, alcohol drinking status, ADL, regular exercise, hear diseases, diabetes, hypertension, digestive system diseases, cerebrovascular diseases and cancer, eye diseases, respiratory diseases, baseline MMSE. Hazard ratios were indicated as solid lines and 95% confidence intervals as grey parts.

**Table 1 nutrients-14-04530-t001:** Characteristics of the 6237 participants at the baseline.

Variables	Total	DDS Change Patterns from Baseline to First Follow-Up					
		High–High	High–Low	Low–High	Low–Low	DDS Change Score	*p*-Value
Number of participants (%)	6237	1700 (27.3)	1092 (17.5)	1255 (20.1)	2190 (35.1)	0.12 ± 3.46	
Age in years, mean (SD)	88.55 (6.93)	87.56 (6.59)	88.50 (6.91)	88.32 (6.66)	89.48 (7.22)	0.12 ± 3.46	
Age group in years							0.027
80–89	3698 (59.3)	1094 (64.4)	642 (58.8)	756 (60.2)	1206 (55.1)	0.20 ± 3.46	
≥90	2539 (40.7)	606 (35.6)	450 (41.2)	499 (39.8)	984 (44.9)	0.07 ± 3.56	
Sex							0.499
Female	3344 (53.6)	734 (43.2)	583 (53.4)	689 (54.9)	1338 (61.1)	0.15 ± 3.45	
Male	2893 (46.4)	966 (56.8)	509 (46.6)	566 (45.1)	852 (38.9)	0.09 ± 3.46	
Type of residence							0.002
Urban	3050 (48.9)	1059 (62.3)	507 (46.4)	627 (50.0)	857 (39.1)	0.26 ± 3.50	
Rural	3187 (51.1)	641 (37.7)	585 (53.6)	628 (50.0)	1333 (60.9)	−0.02 ± 3.41	
Marital status							0.418
In marriage	1495 (24.0)	516 (30.4)	261 (23.9)	286 (22.8)	432 (19.7)	0.18 ± 3.50	
Not in marriage	4742 (76.0)	1184 (69.6)	831 (76.1)	969 (77.2)	1758 (80.3)	0.10 ± 3.45	
Educational background							0.939
Illiteracy	3680 (59.0)	780 (45.9)	635 (58.2)	757 (60.3)	1508 (68.9)	0.11 ± 3.45	
Literacy	2557 (41.0)	920 (54.1)	457 (41.8)	498 (39.7)	682 (31.1)	0.12 ± 3.46	
Living pattern							0.253
With family members	5084 (81.5)	1451 (85.4)	909 (83.2)	1018 (81.1)	1706 (77.9)	0.09 ± 3.46	
Alone or at nursing home	1153 (18.5)	249 (14.6)	183 (16.8)	237 (18.9)	484 (22.1)	0.22 ± 3.43	
Tobacco smoking status							0.045 ^†^
Non-smoker	3933 (63.1)	966 (56.8)	688 (63.0)	799 (63.7)	1480 (67.6)	0.11 ± 3.47	
Current smoker	1261 (20.2)	364 (21.4)	242 (22.2)	232 (18.5)	423 (19.3)	−0.03 ± 3.42	
Former smoker	1043 (16.7)	370 (21.8)	162 (14.8)	224 (17.8)	287 (13.1)	0.33 ± 3.47	
Alcohol drinking status							0.001 ^†^
Non-drinker	4005 (64.2)	998 (58.7)	713 (65.3)	817 (65.1)	1477 (67.4)	0.18 ± 3.46	
Current drinker	1549 (24.8)	500 (29.4)	273 (25.0)	284 (22.6)	492 (22.5)	−0.16 ± 3.42	
Former drinker	683 (11.0)	202 (11.9)	106 (9.7)	154 (12.3)	221 (10.1)	0.37 ± 3.48	
Regular exercise							0.001
Yes	2370 (38.0)	889 (52.3)	441 (40.4)	423 (33.7)	617 (28.2)	−0.07 ± 3.39	
No	3867 (62.0)	811 (47.7)	651 (59.6)	832 (66.3)	1573 (71.8)	0.23 ± 3.50	
Number of teeth, mean (SD)	7.31 (11.82)	8.75 (13.77)	6.68 (10.39)	7.62 (12.38)	6.32 (10.32)	0.12 ± 3.46	
Use of artificial denture	1681 (26.95)	584 (34.35)	301 (27.56)	348 (27.73)	448 (20.46)	0.12 ± 3.46	
BMI, mean (SD), Kg/m^2^	19.38 (5.43)	19.46 (6.15)	19.39 (5.53)	19.44 (5.12)	19.29 (4.94)	0.12 ± 3.46	
Hypertension	898 (14.4)	241 (14.2)	153 (14.0)	176 (14.0)	328 (15.0)	−0.03 ± 3.46	0.158
Diabetes	62 (1.0)	31 (1.8)	9 (0.8)	12 (1.0)	10 (0.5)	0.76 ± 3.71	0.142
Hear disease	438 (7.0)	139 (8.2)	74 (6.8)	76 (6.1)	149 (6.8)	−0.17 ± 3.72	0.069
Cerebrovascular disease	134 (2.1)	37 (2.2)	25 (2.3)	29 (2.3)	43 (2.0)	0.07 ± 3.78	0.888
Digestive disease	217 (3.5)	58 (3.4)	47 (4.3)	34 (2.7)	78 (3.6)	−0.14 ± 3.58	0.270
Cancer	20 (0.3)	7 (0.4)	4 (0.4)	1 (0.1)	8 (0.4)	−1.10 ± 3.63	0.115
Respiratory disease	722 (11.6)	216 (12.7)	122 (11.2)	141 (11.2)	243 (11.1)	0.27 ± 3.48	0.195
Eye diseases	894 (14.3)	254 (14.9)	183 (16.8)	142 (11.3)	315 (14.4)	−0.25 ± 3.42	0.001
Duration of follow-up, months	63.13 (38.71)	68.62 (40.33)	60.53 (37.93)	67.05 (39.64)	57.93 (36.43)	0.12 ± 3.46	

Data were expressed as counts (percentages), except for the age and duration of follow-up; **^†^** Significant difference between current smoker and former smoker (*p* = 0.014); significant difference between the two groups (non-drinker, former drinker) and current drinker, *p* < 0.01.

**Table 2 nutrients-14-04530-t002:** The association between DDS change patterns and incident cognitive impairment.

	Events/Participants	Unadjusted Model		Model 1 ^†^		Model 2 ^‡^	
		HR (95% CI)	*p*-Value	HR (95% CI)	*p*-Value	HR (95% CI)	*p*-Value
DDS change (continuous)	1829/6237	0.96 (0.95, 0.98)	0.000	0.97 (0.96, 0.98)	0.000	0.96 (0.95, 0.98)	0.000
Plant-based DDS change (continuous)	1829/6237	0.96 (0.94, 0.98)	0.000	0.96 (0.95, 0.98)	0.000	0.96 (0.94, 0.97)	0.000
Animal-based DDS change (continuous)	1829/6237	0.95 (0.92, 0.97)	0.000	0.95 (0.93, 0.98)	0.000	0.95 (0.92, 0.98)	0.000
DDS change pattern							
Total DDS							
High–high	355/1700	Reference		Reference		Reference	
High–low	347/1092	1.71 (1.47, 1.98)	0.000	1.56 (1.35, 1.81)	<0.001	1.44 (1.24, 1.67)	<0.001
Low–high	329/1255	1.29 (1.11, 1.50)	0.001	1.18 (1.01, 1.38)	0.028	1.03 (0.88, 1.20)	0.722
Low–low	798/2190	2.04 (1.80, 2.31)	0.000	1.70 (1.50, 1.93)	<0.001	1.43 (1.25, 1.63)	<0.001
Plant-based DDS							
High–high	304/1496	Reference		Reference		Reference	
High–low	353/1157	1.664 (1.427, 1.940)	0.000	1.52 (1.30, 1.77)	<0.001	1.43 (1.23, 1.67)	<0.001
Low–high	325/1217	1.37 (1.17, 1.60)	0.000	1.24 (1.06, 1.46)	<0.001	1.11 (0.95, 1.30)	0.201
Low–low	847/2367	2.05 (1.80, 2.34)	0.000	1.66 (1.46, 1.90)	<0.001	1.44 (1.26, 1.65)	<0.001
Animal-based DDS							
High–high	285/1214	Reference		Reference		Reference	
High–low	294/986	0.65 (0.57, 0.74)	0.000	1.30 (1.10, 1.53)	<0.001	1.18 (1.00, 1.39)	0.047
Low–high	329/1267	0.88 (0.77, 1.00)	0.048	1.11 (0.95, 1.30)	0.205	0.98 (0.84, 1.16)	0.840
Low–low	921/2770	0.72 (0.64, 0.82)	0.000	1.46 (1.28, 1.67)	<0.001	1.22 (1.07, 1.40)	0.004

Model 1 ^†^: Adjusted for age and sex; Model 2 ^‡^: Adjusted for model 1 plus residence, education background, occupation, current marital status, living arrangement, tobacco smoking status, drinking status, regular exercise, number of teeth (continuous), use of artificial dentures, hypertension, diabetes, cerebrovascular diseases, respiratory diseases, digestive system diseases, ADL score, cancer, eye diseases, and BMI (continuous), hear disorders. Other chronic diseases, baseline MMSE.

**Table 3 nutrients-14-04530-t003:** The association between the DDS change patterns and risk of cognitive impairment in subgroups.

Subgroups	Events/Participants		DDS Change Patterns				*p* for Interaction
		DDS Change Score	High–High	High–Low	Low–High	Low–Low	
Age (years)							
80–89	857/3698	0.96 (0.94, 0.98) ^‡^	Ref.	1.66 (1.34, 2.06) ^‡^	1.01 (0.81, 1.26)	1.48 (1.22, 1.80) ^‡^	0.662
≥90	972/2539	0.96 (0.94, 0.98) ^‡^	Ref.	1.25 (1.02, 1.55) ^‡^	1.01 (0.82, 1.25)	1.37 (1.14, 1.63) ^‡^	
Gender							0.854
Male	613/2893	0.97 (0.95, 0.99) ^‡^	Ref.	1.40 (1.09, 1.77) ^‡^	1.02 (0.80, 1.32)	1.37 (1.10, 1.70) ^‡^	
Female	1216/3344	0.96 (0.95, 0.98) ^‡^	Ref.	1.45 (1.20, 1.76) ^‡^	1.02 (0.84, 1.24)	1.44 (1.22, 1.71) ^‡^	
Education							0.948
Illiterate	1309/3680	0.96 (0.94, 0.99) ^‡^	Ref.	1.51 (1.25, 1.83) ^‡^	1.16 (0.96, 1.41)	1.60 (1.36, 1.88) ^‡^	
Literate	520/2557	0.96 (0.95, 0.98) ^‡^	Ref.	1.36 (1.05, 1.75) ^‡^	0.94 (0.72, 1.23)	1.28 (1.02, 1.62)	
Residence							0.016
Urban	738/3050	0.97 (0.95, 0.99) ^‡^	Ref.	1.30 (1.04, 1.63) ^‡^	0.98 (0.79, 1.23)	1.61 (1.33, 1.95) ^‡^	
Rural	1091/3187	0.96 (0.94, 0.98) ^‡^	Ref.	1.55 (1.26, 1.91) ^‡^	1.08 (0.87, 1.33)	1.36 (1.13, 1.64) ^‡^	
Smoking status							0.296
Current or former smoker	545/2304	0.98 (0.96, 1.00)	Ref.	1.29 (0.99, 1.68) ^‡^	1.08 (0.82, 1.41)	1.43 (1.13, 1.81) ^‡^	
Non-smoker	1284/3933	0.96 (0.94, 0.97) ^‡^	Ref.	1.47 (1.22, 1.76) ^‡^	0.99 (0.83, 1.20)	1.41 (1.2, 1.66) ^‡^	
Drinking status							0.117
Current or former drinker	591/2232	0.97 (0.95, 0.99) ^‡^	Ref.	1.59 (1.22, 2.07) ^‡^	1.20 (0.92, 1.56)	1.67 (1.32, 2.12) ^‡^	
Non-drinker	1238/4005	0.96 (0.94, 0.98) ^‡^	Ref.	1.39 (1.15, 1.65) ^‡^	0.95 (0.79, 1.15)	1.33 (1.13, 1.56) ^‡^	
Regular exercise							0.468
Yes	540/2370	0.96 (0.93, 0.98) ^‡^	Ref.	1.42 (1.11, 1.82) ^‡^	1.03 (0.79, 1.35)	1.50 (1.20, 1.87) ^‡^	
No	1289/3867	0.97 (0.95, 0.98) ^‡^	Ref.	1.44 (1.19, 1.74) ^‡^	1.03 (0.86, 1.25)	1.41 (1.20, 1.67) ^‡^	
Living pattern							0.671
Living with family	1505/5084	0.96 (0.95, 0.98) ^‡^	Ref.	1.41 (1.20, 1.66) ^‡^	1.0 (0.85, 1.18)	1.43 (1.24, 1.64) ^‡^	
Living alone	324/1153	0.98 (0.95, 1.01)	Ref.	1.73 (1.14, 2.61) ^‡^	1.31 (0.88, 1.95)	1.58 (1.11, 2.25) ^‡^	

Adjusted for age, sex, residence, education background, occupation, current marital status, living arrangement, tobacco smoking status, drinking status, regular exercise, number of teeth (continuous), use of artificial dentures, hypertension, diabetes, cerebrovascular diseases, respiratory diseases, digestive system diseases, ADL score, cancer, eye diseases, and BMI (continuous), hear disorders. Other chronic diseases, baseline MMSE; ^‡^
*p* < 0.05.

**Table 4 nutrients-14-04530-t004:** HRs (95% CIs) of incident decline in different cognitive domains with the DDS change patterns.

Cognitive Domain	DDS Change Patterns						
	High–High	High–Low		Low–High		Low–Low	
Total DDS		HR (95% CI)	*p*-Value	HR (95% CI)	*p*-Value	HR (95% CI)	*p*-Value
Global	Ref.	1.27 (1.15, 1.40)	0.000	0.93 (0.84, 1.03)	0.200	1.12 (1.03, 1.23)	0.008
Orientation	Ref.	1.23 (1.10, 1.38)	0.000	1.00 (0.89, 1.11)	0.925	1.24 (1.12, 1.36)	0.000
Registration	Ref.	1.19 (1.07, 1.33)	0.001	1.04 (0.94, 1.16)	0.459	1.15 (1.04, 1.26)	0.005
Attention ^†^	Ref.	1.29 (1.16, 1.43)	0.000	1.03 (0.93, 1.15)	0.527	1.22 (1.12, 1.34)	0.000
Memory	Ref.	1.15 (1.04, 1.27)	0.005	0.96 (0.87, 1.06)	0.379	1.05 (0.96, 1.14)	0.291
Language ^‡^	Ref.	1.26 (1.13, 1.39)	0.000	1.00 (0.90, 1.11)	0.981	1.17 (1.07, 1.28)	0.001
Plant-based DDS							
Global	Ref.	1.27 (1.15, 1.40)	0.000	1.06 (0.95, 1.17)	0.294	1.17 (1.07, 1.28)	0.000
Orientation	Ref.	1.24 (1.11, 1.39)	0.000	1.02 (0.91, 1.15)	0.715	1.30 (1.18, 1.44)	0.000
Registration	Ref.	1.15 (1.03, 1.28)	0.015	1.04 (0.93, 1.17)	0.453	1.16 (1.05, 1.28)	0.003
Attention ^†^	Ref.	1.28 (1.15, 1.42)	0.000	1.14 (1.03, 1.27)	0.014	1.25 (1.14, 1.37)	0.000
Memory	Ref.	1.08 (0.97, 1.19)	0.157	1.02 (0.92, 1.12)	0.742	1.06 (0.97, 1.16)	0.213
Language ^‡^	Ref.	1.21 (1.09, 1.34)	0.000	1.03 (0.92, 1.14)	0.633	1.12 (1.02, 1.23)	0.017
Animal-based DDS							
Global	Ref.	1.21 (1.08, 1.34)	0.001	0.94 (0.85, 1.05)	0.260	1.09 (0.99, 1.19)	0.073
Orientation	Ref.	1.05 (0.93, 1.19)	0.444	1.01 (0.89, 1.13)	0.924	1.09 (0.99, 1.21)	0.091
Registration	Ref.	1.12 (0.99, 1.26)	0.077	1.05 (0.93, 1.17)	0.459	1.15 (1.04, 1.27)	0.007
Attention ^†^	Ref.	1.14 (1.02, 1.28)	0.025	1.11 (1.0, 1.24)	0.054	1.15 (1.05, 1.27)	0.004
Memory	Ref.	0.98 (0.87, 1.09)	0.669	1.01 (0.91, 1.12)	0.820	1.02 (0.93, 1.12)	0.614
Language^‡^	Ref.	1.19 (1.06, 1.33)	0.004	1.08 (0.96, 1.20)	0.191	1.16 (1.06, 1.28)	0.002

^†^ Attention and calculation; ^‡^ Language and visuospatial abilities. HRs of decline in different cognitive domains were estimated using Cox proportional hazards models. Adjusting for age, sex, residence, education background, occupation, current marital status, living arrangement, tobacco smoking status, drinking status, regular exercise, number of teeth (continuous), use of artificial dentures, hypertension, diabetes, cerebrovascular diseases, respiratory diseases, digestive system diseases, ADL score, cancer, eye diseases, and BMI (continuous), hear disorders. Other chronic diseases, baseline MMSE.

**Table 5 nutrients-14-04530-t005:** β-Coefficients and 95% CI for the association of the DDS change patterns with MMSE score changes over follow-up time (*n* = 6237). Results from the linear mixed-effects models.

DDS Change Patterns	MMSE Score-β	95% CI	*p*-Value
Baseline			
DDS change categories			
High–high	Ref.		
High–low	−0.403	−0.755, −0.051	0.025
Low–high	−0.202	−0.134, 0.537	0.239
Low–low	−0.286	−0.585, −0.130	0.061
Longitudinal			
High–high × time	Ref.		
High–low × time	−0.157	−0.264, −0.050	0.004
Low–high × time	−0.111	−0.209, −0.014	0.025
Low–low × time	−0.164	−0.252, −0.076	0.000

**Table 6 nutrients-14-04530-t006:** The association between DDS change and the incidence of cognitive impairment.

	Event	Participants	Model 1 ^†^		Model 2 ^‡^	
			HR (95% CI)	*p*-Value	HR (95% CI)	*p*-Value
DDS change (categorical)	1829	6237				
Large decline	216	589	1.39 (1.20, 1.63)	0.000	1.70 (1.44, 2.01)	0.000
Small decline	402	1280	1.11 (0.98, 1.26)	0.100	1.23 (1.08, 1.40)	0.002
Stable status	651	2268	Ref.	Ref.	Ref.	Ref.
Small improvement	397	1483	0.91 (0.80, 1.03)	0.125	0.83 (0.73, 0.94)	0.003
Large improvement	163	617	0.97 (0.82, 1.15)	0.721	0.75 (0.63, 0.90)	0.002

Model 1 ^†^: Adjusted for age, sex; Model 2 ^‡^: Adjusted for model 1 plus residence, education background, occupation, current marital status, living arrangement, tobacco smoking status, drinking status, regular exercise, number of teeth (continuous), use of artificial dentures, hypertension, diabetes, cerebrovascular diseases, respiratory diseases, digestive system diseases, ADL score, cancer, eye diseases, and BMI (continuous), hear disorders. Other chronic diseases, baseline MMSE, baseline DDS.

## Data Availability

Available from the Peking University on request (https://opendata.pku.edu.cn/ (15 October 2021)).

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
