# Peer review of "Adherence to High Dietary Diversity and Incident Cognitive Impairment for the Oldest-Old: A Community-Based, Nationwide Cohort Study"

_nutrients, 2022, doi:10.3390/nu14214530_

Round 1

Reviewer 1 Report

The present study explored the association between changes in dietary diversity and cognitive status in a cohort of older Chinese adults. The study revealed that irrespectively of plant or animal component of the diet, loss of dietary diversity was associated with higher risk of cognitive impairment. The study is clearly well conducted and well presented. I only have some minor points to be discussed in the limitations of the study.

First, the observational nature of the study cannot allow to define cause-effect relationships. Specifically, given the nature of the exposure variable (loss in dietary diversity) and the outcome (cognitive impairment), the authors cannot rule out the possibility for reverse causation, which should be stated in the limitation paragraph.

Second, the methodology used to assess changes in dietary diversity is not really a gold standard and not previously used in other studies (I did not see any reference in the method section). The authors already stated that their data were not quantitative, but they limited this limitation to only the impossibility to calculate the energy intake. I would broad this concept, saying that they attempt to estimate and categorise dietary diversity, but misclassification and lack of validity of this approach should be taken into account when considering the findings from this study.

Minor concerns: please improve Table 6 (perhaps make it large enough as the whole page in order to have a single row for each variable).

Reviewer 2 Report

This is a prospective community-based nationwide cohort Chinese study that focuses on the association between dietary diversity score DDS) and cognitive decline (measured by the MMSE with a cut off score of 18 (because of illiterate participants). Several covariates were considered with appropriate control in statistical models. The association between animal-plant DDS and better cognitive preservations is already well established. The authors attempt to define the novelty of their contribution referring to the focus on 80+ individuals, use of incident cognitive decline (as measures by MMSE changes) instead of binary outcome (decliners or not),long-term follow-up and study of the DDS effect on specific cognitive subdomains. However, most of these parameters have been already considered in the Zheng et al. 2021 study published in Clinical Nutrition (for the same cohort and some common authors). Focus on 80+, 7 follow-ups, use of MMSE changes were already taken into account and the conclusions are very close if not identical. The authors did not comment on this study in Discussion raising doubts about partly redundant publication. In any case, they should discuss in depth this contribution and explain the differences that could merit a new publication.

The methodology is globally sound corresponding to what already published in the previous contributions of the group. There are however some points that need clarification :

This reviewer did not understand why the authors define their high-low DDS groups instead of taking the delta DDS as a continuous variable in their analysis. In fact, their second measurement proposed in Table 3 should be used (and without the distinction between large decline, small decline, etc in Table 6) alone. I don’t see the added value of the ordinal approach that increases tremendously the number of comparisons.

Correction for multiple comparisons is absent. In Table 1, the p values are given without the usual post hoc analysis, The same is true for Table 4 (that in fact is poorly informative since the cognitive subdomain analysis corresponds simply to the MMSE items). Even if the sample is impressive, statistical rigor needs the correction for multiple comparisons. The Table 2 and 3 are presented in a peculiar way. The authors should provide crude data using MMSE changes as the dependent variable and DDS change (continuous), age, gender and other socio-demographic variables as the independent variables (both univariate and multivariable models).

Some results need additional comments : why both high-lowXtime and low-highXtime are negatively associated with MMSE score changes ? What is the percentage of variance of the MMSE score change that is explained by the DDS changes ? Biologically speaking, is it a marginal or a main effect ?

Last but not least, this study just replicates what is already established in the oldest-old.
